# Eligibility traces provide a data-inspired alternative to backpropagation through time

**Guillaume Bellec\*, Franz Scherr\*, Elias Hajek, Darjan Salaj, Anand Subramoney,
Robert Legenstein & Wolfgang Maass**
Institute of Theoretical Computer Science
Graz University of Technology, Austria
{bellec,scherr,salaj,hajek,legenstein,maass}@igi.tugraz.at
\* equal contributions

## Abstract

Learning in recurrent neural networks (RNNs) is most often implemented by gradient descent using backpropagation through time (BPTT), but BPTT does not model accurately how the brain learns. Instead, many experimental results on synaptic plasticity can be summarized as three-factor learning rules involving eligibility traces of the local neural activity and a third factor. We present here eligibility propagation (*e-prop*), a new factorization of the loss gradients in RNNs that fits the framework of three factor learning rules when derived for biophysical spiking neuron models. When tested on the TIMIT speech recognition benchmark, it is competitive with BPTT both for training artificial LSTM networks and spiking RNNs. Further analysis suggests that the diversity of learning signals and the consideration of slow internal neural dynamics are decisive to the learning efficiency of *e-prop*.

### Introduction

The brain seems to be able to solve tasks such as counting, memorizing and reasoning which require efficient temporal processing capabilities. It is natural to model this with recurrent neural networks (RNNs), but their canonical training algorithm called backpropagation through time (BPTT) does not appear to be compatible with learning mechanisms observed in the brain. There, long-term changes of synaptic efficacies depend on the local neural activity. It was found that the precise timing of the electric pulses (i.e. spikes) emitted by the pre- and post-synaptic neurons matters, and these spike-timing dependent plasticity (STDP) changes can be conditioned or modulated by a third factor that is often thought to be a neuromodulator (see [1, 2] for reviews). Looking closely at the relative timing, the third factor affects the plasticity even if it arrives with a delay. This suggests the existence of local mechanisms that retain traces of the recent neural activity during this temporal gap and they are often referred to as *eligibility traces* [2].

To verify whether three factor learning rules can implement functional learning algorithms, researchers have simulated how interesting learnt behaviours can emerge from them [1, 3, 4]. The third factor is often considered as a global signal emitted when a reward is received or predicted, and this alone can solve learning tasks of moderate difficulty, even in RNNs [4]. Yet in feed-forward networks, it was already shown that plausible learning algorithms inspired by backpropagation and resulting in neuron-specific learning signals largely outperform the rules based on a global third factor [5, 6, 7]. This suggests that backpropagation provides important details that are not captured by all three factor learning rules.

Here we aim at a learning algorithm for RNNs that is general and efficient like BPTT but remains plausible. A major plausibility issue of BPTT is that it requires to propagate errors backwards in time

Workshop of the 33rd conference on Neural Information Processing Systems 2019, Vancouver, Canada.

or to store the entire state space trajectory raising questions on how and where this is performed in the brain [8]. We suggest instead a rigorous re-analysis of gradient descent in RNNs that leads to a gradient computation relying on a diversity of learning signals (i.e. neuron-specific third factors) and a few eligibility traces per synapse. We refer to this algorithm as eligibility propagation (*e-prop*). When derived with spiking neurons, *e-prop* fits under the three factor learning rule framework and is qualitatively compatible with experimental data [2]. To test its learning efficiency, we applied *e-prop* to artificial Long Short-Term Memory (LSTM) networks [9], and Long short-term memory Spiking Neural Networks (LSNNs) [10] (spiking RNNs combining short and long realistic time constants). We found that (1) it is competitive with BPTT on the TIMIT speech recognition benchmark, and (2) it can solve nontrivial temporal credit assignment problems with long delays. We are not aware of any comparable achievements with previous three factor learning rules.

Real-time recurrent learning (RTRL) [11] computes the same loss gradients as BPTT in an online fashion but requires many more operations. Eventhough the method is online, one may wonder where can it be implemented in the brain if it requires a machinery bigger than the network itself. Recent works [12, 13, 6] have suggested that eligibility traces can be used to approximate RTRL. This was shown to be feasible if the neurons do not have recurrent connections [6], if the recurrent connections are ignored during learning [12] or if the network dynamics are approximated with a trained estimator [13]. However these algorithms were derived for specific neuron models without long-short term memory, making it harder to tackle challenging RNN benchmark tasks (no machine learning benchmarks were considered in [6, 12]). Other mathematical methods [14, 15], have suggested approximations to RTRL which are compatible with complex neuron models. Yet those methods lead to gradient estimates with a high variance [15] or requiring heavier computations when the network becomes large [14, 11]. This issue was solved in *e-prop*, as the computational and memory costs are the same (up to constant factor) as for running any computation with the RNN. This reduction of the computational load arises from an essential difference between *e-prop* and RTRL: *e-prop* computes the same loss gradients but only propagates forward in time the terms that can be computed locally. This provides a new interpretation of eligibility traces that is mathematically grounded and generalizes to a broad class of RNNs. Our empirical results show that such traces are sufficient to approach the performance of BPTT despite a simplification of the non-local learning signal, but we believe that more complex strategies for computing a learning signals can be combined with *e-prop* to yield even more powerful online algorithms. A separate paper presents one such example to enable one-shot learning in recurrent spiking neural networks [8].

**Eligibility propagation**

**The mathematical basis for *e-prop***    *E-prop* applies for a general class of recurrent network models that includes LSTMs and LSNNs, where each neuron $j$ has a hidden state $\boldsymbol{h}_j \in \mathbb{R}^d$ where $d$ is typically $1$ or $2$ (e.g. the memory cell content of an LSTM unit or the membrane potential for a spiking neuron), and an observable state $z_j^t \in \mathbb{R}$ (e.g. the LSTM outputs or the spikes). The performance of a network on a specific task is usually expressed using a loss function $E$, and learning by gradient descent learning amounts to changing the network weights $\boldsymbol{W}$ such that $E$ is minimized. Much like BPTT, *e-prop* computes the gradient $\frac{dE}{dW_{ji}}$ with respect to the weights from $i$ to $j$ where the neurons $i$ and $j$ are potentially connected. Here, this gradients depends on learning signals $L_j^t$ specific to the neuron $j$ and eligibility traces $e_{ji}^t$ such that (see proof in [16]):

$$\frac{dE}{dW_{ji}} = \sum_t L_j^t e_{ji}^t \ . \tag{1}$$

The eligibility traces $e_{ji}^t$ are defined by $e_{ji}^t = \frac{\partial z_j^t}{\partial \boldsymbol{h}_j^t} \boldsymbol{\epsilon}_{ji}^t$, using so-called eligibility vectors that are expressed recursively and propagated forward in time:

$$\boldsymbol{\epsilon}_{ji}^t = \frac{\partial \boldsymbol{h}_j^t}{\partial \boldsymbol{h}_j^{t-1}} \cdot \boldsymbol{\epsilon}_{ji}^{t-1} + \frac{\partial \boldsymbol{h}_j^t}{\partial W_{ji}} \ . \tag{2}$$

This allows the loss-independent eligibility traces to be defined for any RNN model. For equation (1) to hold, the ideal learning signal is required to be $L_j^t = \frac{dE}{dz_j^t}$. Since this derivative captures how the output $z_j^t$ of neuron $j$ influences the loss $E$ via future observable states of other neurons, its precise value is in general not available at time $t$. For *e-prop*, we replace it by an online approximation, we use

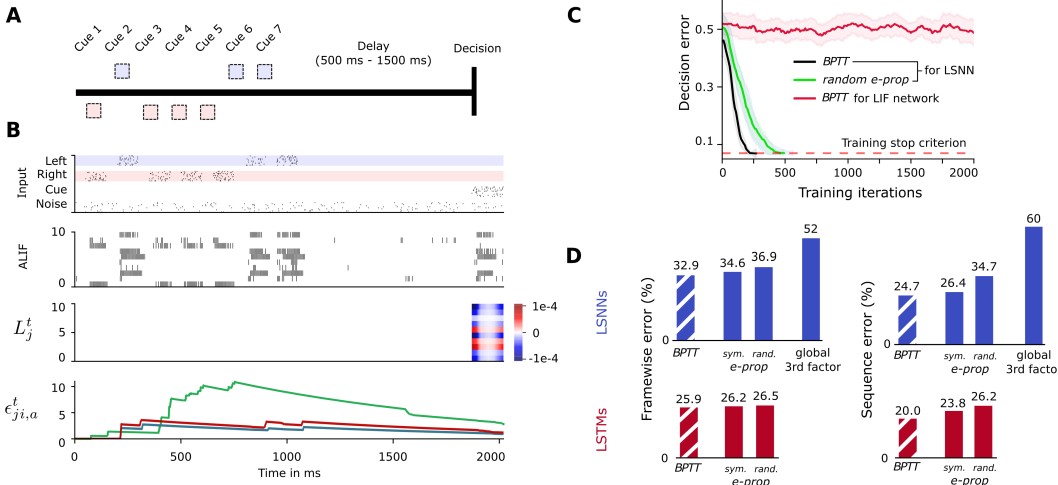

Figure 1: **Solving a task with difficult temporal credit assignment by *e-prop*. A**) Setup of corresponding rodent experiments of [17]. **B**) Input spikes, internal spiking activity of 10 out of 50 sample ALIF neurons, sample learning signals and samples of slow components of eligibility traces in the bottom row. **C**) Learning curves for *BPTT* and *e-prop* for the task in **A**-**B**. **D**) Performance of LSNNs (top) and of LSTMs (bottom) trained with *BPTT*, *symmetric e-prop* and *random e-prop* on frame-wise phoneme classification (left) and on phoneme sequence recognition (right). For LSNNs the performance of *e-prop* with global instead neuron-specific learning signals is also reported.

the partial derivative $\frac{\partial E}{\partial z_j^t}$ which ignores indirect influences through the future activity $\boldsymbol{z}^{t+1}, \cdots \boldsymbol{z}^T$, and only takes into account the direct effect of $z_j^t$ on the loss $E$. For instance, for supervised regression when the network outputs $y_k^t$ are linearly related to the observable states $y_k^t = \sum_j W_{jk}^{\text{out}} z_j^t$, the learning signal becomes $L_j^t = \sum_k W_{jk}^{\text{out}}(y_k^t - y_k^{*,t})$ where $y_k^{*,t}$ are the output targets.

*E-prop* can be implemented online by accumulating the products $L_j e_{ji}^t$ or applying them directly at each time step and does not require to backpropagate through time or to store the past neural activity. This solves a major plausibility issue raised by BPTT. To also avoid the implausible weight sharing between the feedback and feedforward pathways, one can replace the feedback weights in $L_j^t$ by fixed random values as done in [5] leading to the two variants: *symmetric* and *random* e-prop.

**E-prop for spiking neurons and data on synaptic plasticity** To link the theory to data on STDP and three factor learning rules, we applied *e-prop* to a recurrent network of spiking neurons. We use leaky-integrate and fire (LIF) neurons for which the dynamics of the membrane voltage is modelled as a leaky integrator and a spike is emitted when the voltage crosses a firing threshold from below. When simulated in discrete time with a time step of one millisecond, a recurrent network of LIF neurons fits into the general class of RNNs described above such that the hidden state $\boldsymbol{h}_j^t$ is the membrane voltage and the spikes are modelled by a binary observable state $z_j^t \in \{0, 1\}$. Non-differentiability of the binary output of spiking neurons is solved as in [10], by using a pseudo-derivative in-place of $\frac{\partial z_j^t}{\partial \boldsymbol{h}_j^t}$. Remarkably, the eligibility trace $e_{ji}^t$ that emerges for a LIF neuron is a product of the pre- and post-synaptic activity. It is non-zero only if pre-synaptic spikes have preceded a depolarization of the post-synaptic neuron in a time window of about 20 ms which is reminiscent of STDP. Moreover it was verified in [16] that the replacement of $e_{ji}^t$ in equation (1) by a form of voltage dependent STDP used to fit data accurately [2], does not strongly impair the performance of *e-prop* on a pattern generation task [16].

**Forward and stable propagation of RNN gradients with eligibility traces** To enhance the working memory capabilities of the spiking network model, we model slower neural dynamics by introducing a model of firing rate adaptation: after each spike the threshold increases by a constant amount and reduces back to its resting value after hundreds of milliseconds or few seconds. This type of adaptive LIF (ALIF) neuron also includes its current firing threshold in its hidden state $\boldsymbol{h}_j^t$. A

recurrent network of ALIF and LIF neurons connected in an all-to-all fashion is termed LSNN [10]. For an ALIF neuron $j$ each eligibility vector $\epsilon_{ji}^t$ include a slow component $\epsilon_{ji,a}^t$ that decays with the time constants of adaptation, i.e. much slower than for non-adaptive LIF neurons.

We tested the ability of *e-prop* and LSNNs to learn temporal dependencies on a task that was used to study working memory in rodents [17] and that requires to memorize over a delay of hundreds of milliseconds. It requires the rodent to run along a linear track in a virtual environment, where it encounters a number of visual cues to its left and its right, see Figure 1A. After arrival at a T-junction, it has to decide whether it had observed more cues to the left or on the right side and turn towards that direction. This task requires to establish a connection between errors in the decision and the processing of cues that happened a long time ago. We found that LSNNs with 100 neurons can be trained by *e-prop* to solve this task (Figure 1B), but a similar network of LIF neurons without adaptation cannot solve the task (Figure 1C). The key feature arising with adaptive neurons is the slow component $\epsilon_{ji,a}^t$ of the eligibility vector associated with the threshold adaptation and sharing its slow dynamics (see Figure 1B bottom). As the learning signal $L_j^t$ is only non-zero during the decision period at the last time steps of the episode, the eligibility traces must hold the information about the relevant cues for hundreds of time steps during the delay (see Figure 1A). In this way *e-prop* alleviates the need to propagate signals backwards in time.

**Approaching the performance of BPTT**  We compare *E-prop* and BPTT on two benchmarks for RNNs based on the TIMIT dataset: phoneme classification of each audio frame in a spoken sentence, and transcription of the entire sequence of phonemes spoken in a sentence. The LSTM and BPTT baselines were obtained by reproducing the experiments from [18] solving framewise-phoneme classification with 400 LSTM units and [19] solving sentence transcription with three layers of 500 LSTM units. For the LSNN we used 800 spiking neurons in the first task and three layers of 800 spiking neurons in the second. Remarkably, the error rate obtained with *e-prop* is only a few percents larger than the BPTT baseline in all cases, even if the feedback weights are replaced by random ones (Figure 1D). In contrast, the loss performance of *e-prop* when using uniform feedback matrices was significantly worse: the error rate jumped from 34.6 to 52% for frame-wise classification and from 24.7 to 60% for the speech transcription. Beyond supervised learning task it is also shown in [20, 16] that *e-prop* can be applied on reinforcement learning tasks.

**Discussion**  *E-prop* is a novel learning algorithm that qualitatively fits experimental data on synaptic plasticity and maintains a performance that is competitive with BPTT. Our analysis shows that *e-prop* can take advantage of neuron models with enhanced memory capabilities to solve non-trivial temporal credit assignment problems, and the diversity of the learning signals is decisive for the learning efficiency of three factor learning rules. Interestingly, it was found recently that dopaminergic neurons encode more diverse information than a global reward prediction error [17], and performance monitoring neurons providing potential learning signals are found prominently in cortices [21].

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
