# OpenReview forum: "Eligibility traces provide a data-inspired alternative to backpropagation through time"
_NeurIPS.cc/2019/Workshop/Neuro_AI — Real Neurons & Hidden Units @ NeurIPS 2019 Oral_

### Official Review · AnonReviewer3 · 2019-09-20
**A clear and principled solution to an important problem. Missing a more transparent exposition of its limitations and connection to previous literature.**

**Clarity:** 4

**Comment:**

The work presented in this paper is highly relevant to this workshop and a valuable contribution to the field of synaptic plasticity and learning in recurrent networks. There is little question in the mind of this reviewer that this paper merits a high score. That said, in the opinion of this reviewer two important pieces are missing in this paper.

Firstly, the discussion of how the proposed algorithm relates to previous proposals is very limited. In particular, making the explicit connection to real-time recurrent learning (RTRL) is warranted, as these two algorithms are very similar in spirit. Additionally, it seems that e-prop is very similar to the particular RTRL approximation proposed in reference 8. This link also merits further discussion.

Secondly, an interesting question is how the approximations made in e-prop affect its performance. For example, asymptotic performance seems not to be affected (figure D), but learning speed is (figure C). Why is this? And are these limitations inherent to any biologically plausible (i.e. local) approximation to BPTT?

These may have reasonably been omitted due to space constraints, but it would be ideal if they were explored and discussed in the future presentation of this work.

**Category:**

AI->Neuro

**Clarity Comment:**

Mostly well-written and clear.

**Evaluation:**

4: Very good

**Importance:**

5: Astounding importance

**Importance Comment:**

Understanding how synaptic plasticity allows recurrent neural circuits to produce functional patterns of activity is a critical question in neuroscience. This paper directly addresses this question by deriving a synaptic plasticity rule that does exactly this, as well as contextualizing it within a number of related experimental findings.

**Intersection:**

5: Outstanding

**Intersection Comment:**

This paper derives a biologically plausible plasticity rule approximating the backpropagation-through-time (BPTT) algorithm from the artificial intelligence literature, explicitly linking artifical intelligence learning algorithms to biological ones.

**Rigor Comment:**

Due to the space constraints of a 4-page paper, not many mathematical details are provided for the derivation of the proposed algorithm. However, the exposition of the algorithm is clear and principled and the simulations are convincing. One piece that is missing from the results is the limitations of the e-prop algorithm relative to BPTT, given the approximations made in its derivation.

**Technical Rigor:**

4: Very convincing

---

### Official Review · AnonReviewer1 · 2019-09-23
**An exciting step toward biological solutions to temporal credit assignment and gradient computation in recurrent network training**

**Clarity:** 4

**Comment:**

Gives important new results about how eligibility traces can be used to approximate gradients when adequately combined with a learning signal. While eligibility traces have received some attention in neuroscience their relevance to learning has not been thoroughly explored, so this paper makes a welcome contribution that fits well within the workshop goals.

One part that would have been nice to clarify is the relative role of random feedback vs eligibility traces in successful network performance. It also would have been nice to comment on the relationship of this work to unsupervised (e.g. Hebbian-based) learning rules.

A final addition that would have made this work more compelling would have been to more thoroughly explore e-prop for computations that unfold on timescales beyond those built-in to the neurons (e.g. membrane or adaptation timescales) and which instead rely on reverberating network activity.

**Category:**

Common question to both AI & Neuro

**Clarity Comment:**

Given its technical details it was reasonably straightforward to follow.


**Evaluation:**

4: Very good

**Importance:**

4: Very important

**Importance Comment:**

The authors consider how biologically motivated synaptic eligibility traces can be used for backpropagation-like learning, in particular by approximating local gradient computations in recurrent neural networks. This sheds new light on how artificial network algorithms might be implementable by the brain.

**Intersection:**

5: Outstanding

**Intersection Comment:**

The authors directly tried to associate biological learning rules with deep network learning rules in AI.


**Rigor Comment:**

Space is of course limited, but the mathematics presented seem to pass all sanity checks and gives sufficiently rigor to the authors' approach. It would have been nice to present a figure showing how e-prop yields eligibility traces resembling STDP, as this is one of the key connections of this work to biology.

**Technical Rigor:**

4: Very convincing

---

### Official Review · AnonReviewer2 · 2019-09-27
**biologically plausible training of spiking recurrent networks based on slow threshold adaptation**

**Clarity:** 4

**Comment:**

This work is very suitable for the workshop and seems relevant both to machine learning and neuroscience.
Nevertheless, here are a couple of ideas for improvement:
* Relating this work to previous attempts to train spiking neurons would be important. (e.g.
D. Thalmeier, M. Uhlmann, B. Kappen, and R.-M. Memmesheimer 2016,  DePasquale, B., Churchland, M.M. & Abbott, L.F. 2016, R. Guetig 2016, Kim, Chow 2018, A. Ingrosso, L.F. Abbott 2018)
* How does the network capacity scale with network size?
* Is the low irregularity (coefficient of variation of inter-spike intervals after training seems very small) a feature of the learning algorithm? If yes, how can this get more realistic irregular?
* What is the dynamic state of networks after training? Is there a cancellation of external inputs by net inhibitory recurrent interaction, like in a balanced state? How do pairwise correlations change during training and are they biologically plausible?
* Are spikes in this framework necessary for computation, or are they just a biologically plausible feature that doesn't harm too much? If spikes are not required, could this be mapped to a rate-based analogous network e.g. with BCM-like plasticity, where analytical results might be easier to achieve?
* What are the core mechanisms of this learning algorithm and how could they be understood in more detail?
* How could this be used to implement reinforcement learning, regression, classification, time-series prediction?
* (How) Can E-prop be characterized analytically in a simplified form/on a toy problem?
* Which experimentally testable predictions arising from this work?

**Category:**

Common question to both AI & Neuro

**Clarity Comment:**

The problem is stated clearly, the methods are explained well. Because of the limited space, the derivation is only conceptually understandable not in every mathematical step, but the reviewer can't blame the authors for that. The results are clear and understandable.


**Evaluation:**

5: Excellent

**Importance:**

5: Astounding importance

**Importance Comment:**

This work addresses how temporal credit assignment can be solved in spiking recurrent networks. Based on approximate gradients of a loss in recurrent spiking networks with threshold adaptation, a biologically plausible local learning rule is derived that involves an eligibility trace, pre- and postsynaptic activity.  The results seem unparalleled both in terms of performance and biological plausibility and open a promising avenue to implement (reinforcement) learning in spiking neural networks.

**Intersection:**

5: Outstanding

**Intersection Comment:**

The problem of credit assignment in recurrent networks is relevant both for machine learning and for neuroscience. While superficially, this works seems mostly to be a biologically plausible implementation of gradient-based learning in recurrent spiking networks, it might also provide inspiration for the machine learning community to think beyond discrete-time firing RNN. Currently spiking networks are barely used in machine learning despite their advantages (e.g. lower energy need), because it seems difficult to train them to do something useful, hopefully, this paper might be a step towards changing this.

**Rigor Comment:**

The derivation of a local and biologically plausible learning rule is only partially understandable (because of the limitations of the 4-page format), but the general concept is clear. It is, however not clear how different simplifying assumption in the approximation of the gradients are justified and why they have only a minor effect on the final performance. Moreover, the robustness of the results with respect to details of the parameters is not apparent. Is the excellent performance only observed in a small parameter regime that requires fine-tuning, or is it a general feature? When does it break down and why? Is the assumption of a fully connected network crucial, or would this also work on sparse networks?

**Technical Rigor:**

4: Very convincing

---

### Decision · Program_Chairs · 2019-10-02

Accept (Oral)